# Fungal Laccases: Fundamentals, Engineering and Classification Update

**DOI:** 10.3390/biom13121716

**Published:** 2023-11-28

**Authors:** Pablo Aza, Susana Camarero

**Affiliations:** Margarita Salas Center for Biological Research, Consejo Superior de Investigaciones Científicas (CSIC), 28040 Madrid, Spain; pabloaza@cib.csic.es

**Keywords:** multicopper oxidases, laccases, basidiomycete fungi, catalytic activity, applications, directed evolution, classification

## Abstract

Multicopper oxidases (MCOs) share a common catalytic mechanism of activation by oxygen and cupredoxin-like folding, along with some common structural determinants. Laccases constitute the largest group of MCOs, with fungal laccases having the greatest biotechnological applicability due to their superior ability to oxidize a wide range of aromatic compounds and lignin, which is enhanced in the presence of redox mediators. The adaptation of these versatile enzymes to specific application processes can be achieved through the directed evolution of the recombinant enzymes. On the other hand, their substrate versatility and the low sequence homology among laccases make their exact classification difficult. Many of the ever-increasing amounts of MCO entries from fungal genomes are automatically (and often wrongly) annotated as laccases. In a recent comparative genomic study of 52 basidiomycete fungi, MCO classification was revised based on their phylogeny. The enzymes clustered according to common structural motifs and theoretical activities, revealing three novel groups of laccase-like enzymes. This review provides an overview of the structure, catalytic activity, and oxidative mechanism of fungal laccases and how their biotechnological potential as biocatalysts in industry can be greatly enhanced by protein engineering. Finally, recent information on newly identified MCOs with laccase-like activity is included.

## 1. Laccases: General Aspects

Laccases, EC 1.10.3.2, are oxidoreductase enzymes with polyphenol oxidase activity that belong to the multicopper oxidase superfamily (MCO), which also includes ascorbate oxidases (EC 1.10.3.3) and ferroxidases (EC 1.16.3.1), among others. Laccase activity depends on several catalytic copper ions for the oxidation of the substrate and the reduction of O_2_ to H_2_O as a by-product of the catalysis [1]. Originally discovered in the exudates of the oriental lacquer tree *Toxicodendron vernicifluum* (formerly *Rhus vernicifera*) [2], laccases have been identified in fungi [3], bacteria [4] or even insects [5], making them one of the most ubiquitous enzymes in nature. Laccases are involved in several physiological roles. In bacteria, they contribute to pigment synthesis, sporulation and protection against oxidative stress and UV light, while plant laccases are implicated in cell wall lignification, as well as in damage and stress response [6]. In insects, laccases participate in cuticle sclerotization and pigmentation [5,6]. As for fungal laccases, in addition to their involvement in various functions such as defense/protection, virulence or pigment formation, their most studied role is in lignin biodegradation [7,8]. In fact, fungal laccases have traditionally been much more widely studied than their counterparts, particularly those secreted by white-rot basidiomycetes [9]. Some of these laccases have the highest redox potentials described so far [10,11], which gives them superior oxidation capacities on a wider range of substrates and, thus, a higher applicability as biocatalysts. With this in mind, this review focuses on fungal laccases, with special emphasis on basidiomycete laccases. We give an overview of their structure, catalytic properties and biotechnological applicability, particularly addressing their heterologous expression and engineering to obtain tailor-made biocatalysts, and provide the recent advances made by our group in the classification and characterization of enzymes with laccase-like activity.

### 1.1. Structure

Fungal laccases are mostly extracellular glycoproteins of approximately 60–70 kDa with diverse carbohydrate moieties whose content typically varies from 10% to 25%, although higher saccharide contents have also been reported [12]. Alike other MCOs, they are typically monomeric enzymes whose polypeptide chain folds in three cupredoxin-type domains (D1, D2 and D3) formed by β-sheets arranged in a classical Greek-key barrel structure (Figure 1). Two conserved disulphide bridges stabilize the folding, connecting D1 with D2 and D1 with D3 [13,14]. The functional unit of fungal laccases includes four catalytic coppers covalently coordinated to the protein backbone by ten histidine residues and one cysteine residue [7]. 

The catalytic copper atoms are classified based on their spectroscopic characteristics [16,17]:Type 1 copper site (T1): located in D3, it is coordinated by two histidine residues and one cysteine residue in a trigonal coplanar arrangement. The Cu-S(Cys) bond is responsible for the typical blue color associated with these enzymes, resulting in a pronounced absorption in the visible region at 600 nm and a small parallel hyperfine coupling constant in electron paramagnetic resonance (EPR).Type 2 copper site (T2): composed of one copper coordinated by two histidines, it shows no absorption in the visible spectrum but reveals paramagnetic properties.Type 3 copper site (T3): this site is a binuclear center with two catalytic coppers coordinated by six histidine residues (three for each T3 copper atom). It is spectroscopically characterized by absorption at 330 nm and the absence of an EPR signal due to the antiferromagnetic coupling of the copper pair.

The T2 and T3 coppers together form the trinuclear cluster (TNC), connected to the T1 site via a conserved His-Cys-His triad (two histidine ligands of the T3 coppers and the cysteine ligand of the T1 copper) [18].

The binding pocket, situated near the catalytic T1 site, is determined by several flexible loops that vary in size, shape and amino acid composition among laccases. This amino acid diversity is important for determining the protein–substrate interactions that ultimately define the broad oxidation capabilities of these enzymes [15,19,20,21].

### 1.2. Catalytic Activity

Fungal laccases exhibit a broad substrate spectrum. They catalyze the oxidation of several organic compounds like o- and p-substituted phenols and aryl amines, N-heterocycles, or different synthetic organic dyes [22,23,24]. These substrates undergo monovalent oxidation at the T1 site, and electrons are swiftly transferred via the His-Cys-His triad pathway to the TNC, where O_2_ is reduced to form two H_2_O molecules (Figure 2). 

The O_2_ reduction mechanism has been studied as common to all MCO members. The catalytic cycle starts with the four catalytic coppers fully oxidized in a resting state. In this state, the T3 coppers are hydroxide-bridged, and the T2 copper is bonded with an external hydroxo ligand [17]. As substrate oxidation occurs at the T1 site, electrons are transferred internally until all four catalytic coppers are fully reduced, reaching the reduced stage (Figure 3). In the next catalytic step, the TNC reacts with O_2,_ and the copper ions are oxidized two by two, in two consecutive steps. First, a peroxide intermediate is formed with two oxidized coppers (T2 and one T3). Next, the reductive cleavage of the O–O bond occurs via the transfer of two electrons from the remaining reduced coppers (one T3 and T1), assisted by proton donation from carboxylate residues near the TNC, to form the native intermediate. This state quickly reverts to the fully reduced form by the oxidation of four additional substrate molecules, while in the absence of more molecules of reductants, the native intermediate slowly decays to the resting oxidized form [17]. Three conserved acidic residues assist the catalysis. An aspartic acid inside the binding pocket (Asp205 according to PM1 laccase numbering, [15]) participates together with the His455 ligand in the concerted electron-proton transfer for the oxidation of phenols at the T1 site [26], while two conserved aspartates near the TNC (Asp77 and Asp453, PM1 laccase numbering) act as proton donors during the reductive cleavage of peroxide [17]. 

In laccases, electron transfer to T1 and TNC is governed by an inner and second sphere of residues surrounding the T1, i.e., the copper-binding residues and several amino acids not directly bound to T1 copper, respectively. Both are known to modulate the oxidative properties [27]. Although this MCO catalytic oxidation is widely adopted, a recent study has suggested a scheme of oxygen reduction in the TNC that differs in some details. This alternative model differs in the symmetry of the peroxide’s locations, the positions of the two oxygen ligands, and the sequence of oxidation for the T2 and T3 ions. The variations also extend to details of substrate binding and the release of products from the TNC [28].

The reduction potential (*E*°) of the T1 site determines the oxidation efficiency on substrates with high ionization potentials [29]. Based on this, laccases may be divided into low (*E*° < 500 mV), medium (*E*° 500 to around 700 mV), and high (from >720 mV) redox potential enzymes; the latter are mostly isolated from basidiomycete fungi [30]. In low-redox-potential laccases, the reduction of T1 copper by the substrate is the rate-limiting step of the reaction [17]. By contrast, the higher redox potential of T1 in fungal laccases decreases the rate of the intramolecular electron transfer (IET) for the reduction of the native intermediate to the fully oxidized form. However, the IET is faster enough compared with the decay rate of the native intermediate, which makes this state catalytically relevant, enabling a fast turnover [31]. 

Despite the extensive knowledge about these enzymes, the structural determinants that modulate their redox potential are yet not completely elucidated. The most accepted significant contribution to the T1 copper *E*° is the presence or absence of the fourth axial ligand, which produces a perturbation of the geometry of this site [32,33]. In plant and bacterial laccases, a methionine occupies this position, providing a weak bond with the T1 copper, resulting in a distorted tetrahedral geometry associated with low-redox-potential laccases. Conversely, in fungal laccases, the axial position is mostly occupied by a non-coordinating phenylalanine or leucine residue giving rise to a trigonal planar geometry associated with laccases with a higher redox potential [16,34,35].

Other factors suggested to modulate the *E*° are not necessarily ascribed to the nature of the T1 copper ligands. For instance, side chains of non-polar residues located in the loops delineating the substrate cavity or in close proximity to T1 copper provide a hydrophobic environment important for tuning the *E*° of laccases [36]. Similarly, non-covalent interactions, such as hydrogen bonding networks and dipole environment near T1 copper, or interacting directly with T1 ligands, influence their conformation and the *E*° value [36,37]. Interactions that affect the positioning of the α-helix containing the His 455 ligand (PM1L numbering, Figure 2 [15]) can cause an elongation of the coordination distance between the T1 copper and His455, potentially increasing the electron deficiency of T1 and therefore increasing the value of the *E*° [35,38]. 

### 1.3. Redox Mediators

The oxidative capabilities of laccases can be enlarged in the presence of certain low-molecular-weight compounds that act as redox mediators. These compounds serve as laccase substrates, and their oxidation by the enzyme generates diffusible radicals. These radicals are capable of oxidizing other molecules that are recalcitrant to direct oxidation by the enzyme due to their high-redox-potential (Figure 4) or complex substrates that are not readily accessible to the enzyme binding pocket [39,40]. After the first description of ABTS as a mediator of laccases [41], other efficient artificial mediators have been deemed those generating nitroxyl radicals such as 1-hydroxybenzotriazole (HBT) [42], violuric acid [43] or 2,2,6,6-tetramethyl-1-piperidinyloxy free radical (TEMPO) [44]. The combination of laccases with these molecules in the so-called “laccase-mediator systems” has proved to efficiently enlarge the substrate repertory of the enzymes and their efficiency for the oxidation of recalcitrant molecules or complex polymers [45,46,47,48]. However, despite their demonstrated advantages, the high cost of some of these mediators and the potential release of toxic derivatives are drawbacks to their application. In this regard, certain natural phenolic compounds, derived from lignin biodegradation or fungal metabolism, can act as laccase mediators. Thus, they represent a sustainable alternative to the aforesaid artificial compounds [49,50,51]. For instance, certain lignin-derived phenols that are easily obtained from industrial biomass waste [52] have been shown to be effective in the oxidation of recalcitrant synthetic organic dyes [53], polycyclic aromatic hydrocarbons [54], or lignin and lipids in paper pulps [52]. 

## 2. Multicopper Oxidases Reclassification

Laccases are the largest and widest distributed group of MCOs, with the most diverse functions in nature. As regards fungal laccases, they share conserved structural determinants and a common capability to oxidize aromatic compounds and lignin [55,56]. Several studies have been focused on identifying distinctive and concise ways for classifying them—for instance, by grouping laccases according to specific protein residues such as the amino acid occupying the axial ligand position [57], as well as by differentiating them according to their substrate specificity [58]. Sequence homology-based approaches have shed light on the common structural features of this MCO family. Kumar and co-workers suggested the first amino acid signature sequences for distinguishing laccases from other MCOs, reporting a total of four motifs (L1–L4) that comprise the copper ligands and contiguous residues [55]. Later on, a more precise classification of laccases and other MCO families was redrawn based on phylogenetic analysis [59]. 

The phylogeny of fungal laccases has been addressed in different studies revealing the genetic complexity of these enzymes, with multiple laccase genes evolved through duplication-divergence events [60] or the suggestion of clustering patterns with respect to enzyme properties [61]. These efforts have contributed enormously to a better understanding of laccases, but their overlapping substrate specificity and poor sequence homology still make an accurate classification difficult. In addition, continuous advancements in massive genome sequencing are yielding new enzyme sequences, with many new MCO entries automatically annotated as laccases that, in most cases, still await experimental verification.

On the other hand, the knowledge on fungal laccases has been, for many years, biased to those enzymes encoded by white-rot basidiomycetes from the Polyporales order due to their role in lignin biodegradation during wood decay [9]. In this regard, the recent study of Savinova and co-workers provides a detailed evolutionary analysis focused on Polyporales laccases. They suggest a common single ancestral gene for all Polyporales laccases, which have evolved from this gene via extensive duplications, in parallel with the evolution of angiosperms [62]. The study of other basidiomycete orders such as Agaricales, Russulales, or Boletales has been relegated to second place, even when these fungi can colonize different lignocellulosic materials, constituting valuable sources of laccases and other types of MCOs. In a recent comparative genomic study of 52 basidiomycete fungi from various orders and with diverse lifestyles, we made a revisited classification of MCOs in an attempt to better understand the role of laccases in lignocellulose degradation and the distinctive features of laccase-like enzymes respecting other MCO members [63]. The phylogenetic analysis revealed a total of 649 MCO enzymes assembled in different clusters according to their conserved structural motifs and theoretical activities as: Ascorbate Oxidase (AO), Ferroxidase (FOX), Laccase-Ferroxidase (LAC-FOX) and Laccase sensu stricto (LAC). In addition, three novel clusters of laccase-like enzymes separated but related to laccases sensu stricto were described as: Novel Laccase (NLAC), Novel MCO (NMCO) and Novel Laccase with potential ferroxidase activity (NLAC-FOX) (Figure 5). The more relevant properties of the different MCO groups are described below. 

**Ascorbate Oxidases (AOs).** They were scarcely found in the 52 fungal genomes studied. Only four of them harbored AO genes (only one per genome), with the exception of *Schizophyllum commune* [63]. AO enzymes typically catalyze the oxidation of ascorbic acid to dehydroascorbate [64]. However, their substrate specificity might not be that restricted, since their activity towards phenolic substrates has been reported [65]. In nature, AOs seem to influence growth and regulate the redox state [66]. Structurally, these enzymes lack disulphide bonds that connect and stabilize crupredoxin domains in other MCOs. They have a methionine in the position of the fourth axial ligand of T1 copper and a leucine instead of the commonly conserved aspartic acid in the 205th position in laccases (PM1L numbering, PDB 5ANH).

**Ferroxidases (FOXs).** These enzymes were identified in every species of the 52 fungal genomes studied, having on average one FOX gene per genome, except for three genomes which had none [63]. Their natural activity is related to iron uptake and metal homeostasis [67]. Acquisition of ferrous ion oxidation in these MCOs relies on the presence of a specialized Fe (II)-binding site composed of three acidic residues equivalent to the Glu185, Asp283 and Asp409 of the extensively studied ferroxidase Fet3p from *S. cerevisiae*. Additionally, in Fet3p, Glu185 and Asp409 are part of the electrical wire that connects Fe (II) to T1 copper by means of hydrogen bonds with the two histidine ligands of T1, constituting a pathway for electron transfer from the iron to the T1 site [68,69].

**Laccase-Ferroxidases (LAC-FOXs).** MCO enzymes with hybrid laccase and ferroxidase activities have been identified and characterized in the Tremellomycetes *Cryptococcus neoformans* fungus [70,71], and in Agaricomycetes belonging to Polyporales species, such as *Phanerochaete chrysosporium* and *Phanerochaete flavido-alba* [19,72] or Russulales, e.g., *Heterobasidion annosum s. l.* [73]. This dual activity relies on the presence of some of the aforesaid catalytic determinants (Glu185, Asp283 and Asp409) allowing ferroxidase activity in ferroxidase Fet3p from *S. cerevisiae* [68,69]. For instance, the MCO1 laccase from *P. chrysosporium* harbors two acidic residues equivalent to Glu185 and Asp409. The enzyme has been proven to show efficient ferroxidase activity similar to that of Fet3p, while it oxidizes typical laccase substrates like aromatic amines, ABTS and phenols [19]. Additionally, the *P. flavido-alba* enzyme, holding only the equivalent residue to Asp183 [73], exhibited ferroxidase activity together with laccase activity on aryl amines and phenols [72]. Similarly, the MCO of *C. neoformans* showed also hybrid activity [70,71], although it only has the equivalent to Asp409 [73]. By contrast, a novel LAC-FOX from *Heterobasidion annosum s. l.* displayed good laccase activity but no Fe (II) oxidation activity, despite holding the residue equivalent to Asp409. This LAC-FOX enzyme only displayed ferroxidase activity after the full restoration of the three acidic determinants. The variant also retained activity on a broad spectrum of laccase substrates [73].

The LAC-FOX cluster was already described in previous phylogenetic studies [59], and later on confirmed by Ruiz-Dueñas and co-workers [63]. A more recent study suggested this MCO family may be subdivided into two differentiated phylogenetic subgroups based on the presence/absence of some of the acidic determinants. One subgroup is comprised of enzymes that conserve equivalents to Glu185 and Asp409 residues and show efficient ferroxidase activity, while proteins from the second group only show the equivalent to Asp409 [73].

In the study of 52 genomes, it was concluded LAC-FOX enzymes mainly appeared in wood-rotting fungi. One LAC-FOX was found on average in every white-rot and brown-rot species regardless of whether they belong to Polyporales, Agaricales or Boletales [63]. Structurally, they have a conserved leucine in the position of the fourth axial ligand of the T1 site, one disulphide bond and normally a phenylalanine replacing the acidic residue at the 205th position. The carboxylic side-chain of the acidic residue is considered to assist the deprotonation of phenolic compounds during their oxidation [26], and it has been related as well to facilitate the substrate binding of amines [74].

**Laccases sensu stricto (LAC).** This group constitutes the largest cluster of MCOs, with a total of 465 sequences out of the 649 MCOs found in the 52 fungal genomes [63]. In general, the highest number of laccases were found in forest-litter degrading species (around 20 laccase genes in several species). This trend might be related to the more complex and heterogeneous substrates that forest-litter degrading fungi living in soil have to face. Moreover, some wood decayers belonging to white-rot species also had an important number of laccases, whereas brown-rot basidiomycetes presented a significantly reduced number of laccases. Most of the enzyme sequences had a phenylalanine or leucine residue occupying the putative fourth axial ligand of T1 Cu, a typical feature of laccases with higher redox potential [16,34,35]. Most laccases conserved the typical acidic residue of Polyporales laccases in the 205th position (PM1L numbering, PDB 5ANH), although some exceptions were found in laccases from Agaricales, Boletales or Russulales orders. Finally, all laccases kept the two disulphide bonds typical of canonical laccases.

**Novel Laccases (NLACs).** In the 52-fungal-genome study, up to 28 laccase sequences from different species were grouped in a separated cluster from laccases sensu stricto, with 1-2 proteins from different species gathering together. The NLACs were confined to Agaricales and Russulales species, whereas none were identified in the Polyporales species [63]. As signature structural characteristics of NLACs, they hold a conserved leucine in the position of the fourth axial ligand of T1 copper and exhibit an arginine residue instead of the conserved aspartic acid at the 205th position found in LAC. The amino-acid residues delimiting the substrate-binding pocket in NLAC enzymes also differ considerably from laccases sensu stricto.

Laccases sensu stricto are normally monomeric enzymes [13,14], whereas identified NLACs form heterodimers with small proteins of unknown function. Interestingly, all fungal genomes with a classified NLAC also had at least one gene encoding a small protein, which could indicate the formation of a putative heterodimer [63].

The contribution of the small subunit to increase the stability of the enzyme in the heterodimer has been suggested in POXA3 [75]. An equivalent heterodimeric complex of another member from *P. eryngii var. ferulae* with a small protein showed the enhancement of enzyme stability and a superior catalytic activity [76]. This has been recently confirmed during the expression and characterization of the heterodimeric complex formed by the NLAC of *P. eryngii* with a small protein identified in the genome of the fungus. In that study, the stability to pH, temperature and presence of organic co-solvents and the catalytic activity of the enzyme was remarkably increased by the presence of the small subunit [77]. Moreover, the crystal structure of a small subunit was solved for the first time. Finally, the observed interactions between the catalytic and the small subunit indicated that the NLAC holds structural features likely involved in substrate binding and/or the interaction with the small subunit, which could explain the differences in activity and stability of monomeric or complexed NLACs, as well as of NLACs compared to laccases sensu stricto [77].

**New Multicopper Oxidases (NMCOs).** In the 52-fungal-genome study, up to 29 atypical MCO sequences segregate from the rest of laccase-like enzymes in a separate cluster named NMCO [63]. These enzymes were only found in a few Agaricales and Russulales species. All were characterized by the absence of three of the ten conserved histidine residues that coordinate the catalytic coppers of the TNC in all MCOs, which were replaced by basic or acidic amino acids. In addition, and like NLACs, many NMCOs show an arginine residue instead of the conserved aspartic acid in position 205 and exhibit a binding pocket with more acidic amino acids exposed to the solvent than laccases sensu stricto [63].

**New Laccase-Ferroxidases (NLAC-FOXs).** In the 52-fungal-genome study, certain species of Agaricales harbor some laccase-like sequences that are grouped separately in the phylogenetic tree. This group was named NLAC-FOX due to the presence of one or two of the acidic residues needed for Fe (II) binding and oxidation [63]. These sequences have a conserved leucine in the position of the fourth axial ligand of T1 Cu, a tyrosine replacing the conserved acidic residue at the 205th position of laccases sensu stricto, and a sole disulphide bond. As for NMCOs, the relevance of these distinct features in the catalytic activity and physiological role of NLAC-FOX awaits experimental validation.

## 3. Biotechnological Applications 

Fungal laccases possess three main properties that make them versatile biocatalysts for the industry: non-specific catalytic activity on a range of aromatic compounds and lignin polymers (whether alone or in the presence of redox-mediators); they are eco-friendly (they only require oxygen and produce water); and, in some cases, they possess high redox potential. Consequently, they stand out as oxidoreductases with the largest number of reported applications to date in different sectors (Figure 6).

**Pulp and paper industry.** Most of the lignin is removed during the cooking of wood chips to obtain cellulosic pulp. The residual lignin remaining in the crude pulp is subsequently removed through bleaching sequences with chlorine-derived reagents to produce bleached paper pulps. Today, the use of chlorine dioxide (ClO_2_) as a bleaching agent in modern paper mills can be reduced by the application of enzymes such as laccases and xylanases [78,79,80,81]. Laccases, in the presence of redox mediators, can be used in a pre-bleaching step for removing the color caused by lignin, allowing us to reduce the use of bleaching chemicals with the additional benefit that cellulose is not degraded during the process [11,42,80]. In addition, laccases can be applied for deinking recycled paper, thus reducing the environmental impact [82]. Laccase-mediator systems have been also used to control pitch and lipidic deposits that reduce the quality of the pulps and cause problems in the mill circuits [83]. In addition to this, laccases can be used for valorizing technical lignins—the by-products from this industry—through the functionalization or grafting of fibers with target compounds to make added-value materials [80,84], or through lignin depolymerization to obtain lignin-derived compounds that can be used as components of bio-based polymers or materials [85].

**Textile**. In a similar approach to the pulp and paper industry, laccases catalyze the decolourization of textile organic dyes so that they can be used in finishing denim fabric. Given their oxidative activity towards indigo dyes, fungal laccases can provide the characteristic worn or stonewashed effect to denim fabrics by the partial removal of the indigo dye, as demonstrated with a laccase from *Trametes versicolor* [86]. On the other hand, fungal laccases provide an environmentally friendly synthesis of novel organic dyes and fabric-dyeing technologies by catalyzing the oxidative coupling of aryl amines and phenols, as demonstrated with a variant of PM1 laccase engineered by our group [10] or with *Pleurotus ostreatus* laccase [87]. Furthermore, laccases can add novel characteristics to the fabrics, like conductivity [88], or antimicrobial properties through the enzymatic coating of fibers [89].

**Bioremediation**. Laccases can meet demanding environmental regulations for wastewater treatment from different industries due to their ability to oxidize a wide range of aromatic pollutants into less toxic derivatives. Chlorine-based compounds, phenols or polycyclic aromatic hydrocarbons can be transformed by these enzymes into intermediates more amendable to secondary treatments [54,90,91]. In this context, laccases have been successfully evaluated for the biodegradation of pesticides such as chlorophen and dichlorophen from water [92,93,94]. Laccase-mediator systems can be a green alternative for the treatment of wastewaters polluted with azo dye-derived products from textile or paper printing industries [53]. Moreover, due to their natural activity towards phenols, these enzymes can contribute to reducing the pollution of high-phenolic-content wastewater from olive-oil mills and distilleries [95]. As for pharmaceuticals and personal care products, laccases have been studied for eliminating emerging antibiotic pollutants from wastewaters [96] or other emerging contaminants like naproxene and carbamazepine [97], together with endocrine-disrupting agents, steroid hormones, and microplastics [98].

**Organic synthesis**. Laccases have also drawn attention in organic synthesis due to their ability to catalyze the oxidative coupling and dimerization of several compounds with potential industrial applications, avoiding toxic reagents and chemical synthesis. As aforementioned, several synthetic organic dyes used in paper printing and the textile industry [10,99] can be obtained using laccase as a biocatalyst. Fungal laccases catalyze the synthesis of organic dyes through the oxidative polymerization of phenolic compounds [100] or aryl amines with phenols. The coupling of phenylenediamine and α-naphtol catalyzed by *P. ostreatus* POXA1b laccase renders SIC-RED dye, while the coupling of resorcinol and 2,5–diaminobenzenesulfonic acid renders other colored compounds [87,101]. Fungal laccases also catalyze the synthesis of polymers such as polycatechol, a polymer used in chromatographic resins and biosensors [102], or polyaniline [103], an electro-conductive polymer with many applications [104,105]. Furthermore, laccases can be of interest to the pharmaceutical sector as biocatalysts of the synthesis of relevant medical products, such as antitumoral drugs like actinocin, which is obtained by the oxidation of 4-methyl-3-hydroxyanthranilic acid [106,107] or vinblastine [108], or to conjugate catechins and dextran to create anticancer drugs, as well as generate new derivatives from resveratrol and β-estradiol [109,110,111]. Recently, these enzymes have been proposed to selectively couple phenols as a tailoring step in polyketide synthesis—a rich source of pharmaceutical and agrochemical lead compounds [112].

**Food Industry**. Laccases are used in beverage processing to remove phenolic compounds and enhance or stabilize the organoleptic properties of the final products as an alternative to chemical methods—for instance, to eliminate aromatic compounds, prevent the loss of flavor and color quality in wine and prolong beer half-life [95,113]. Additionally, in the bread-making process, laccases modify bread texture and dough consistency, increasing the strength and stability of the final product [95,114].

## 4. Laccase Engineering and Heterologous Production

Fungal laccases are biocatalysts of interest for industrial purposes, as evidenced by the applications mentioned in the previous section and patents filled [115]. However, the harsh operational requirements of the industrial processes (e.g., high temperature, extreme pH, ionic strength, etc.) often preclude the integration of wild-type enzymes. In silico screening of genomes and databases allows for the discovery of wild-type enzymes from extremophiles with potential properties of interest under harsh industrial conditions [116]. Alternatively, it is possible to endow native enzymes with new functionalities under non-natural (extreme) conditions by using protein engineering to adapt the enzymes to the target industrial process [11,117]. 

In this line of interest, the directed evolution of enzymes, pioneered by Frances Arnold in the early 90s, arose as a powerful alternative to rational design in order to adapt enzymes to industrial requirements [118].

### 4.1. Engineering of Fungal Laccases

Directed evolution reduces to practice the main processes of Darwinian evolution at the molecular level and has a scale of weeks or months of work in the laboratory. Through iterative rounds of genetic diversification and selection, the accumulation of beneficial mutations in the protein sequence enables the in vitro evolution of the enzyme towards desired traits, such as improved catalytic properties under non-natural conditions [119], or even novel functionalities not found in nature [120]. Today, enzyme-directed evolution has become an essential part of biotechnological industries to design tailor-made biocatalysts, and it is still under continuous development, with recent advances in library design [121,122], methods of (ultra)high-throughput screening [123], or in vivo continuous evolution [124]. Furthermore, directed evolution constitutes an invaluable tool for elucidating evolutionary principles [125].

A typical directed evolution cycle comprises three steps: (i) genetic diversification of the DNA encoding the starting enzyme, (ii) expression of thousands of enzyme variants in active form, and (iii) screening of the library in a high-throughput fashion to quickly identify those mutants that exhibit improvements on the targeted property (Figure 7). The fittest mutant(s) serve as a template for the next round of diversification and selection, and the process is repeated as many times as required until the desired level of improvement is achieved [126].

Several studies in the literature illustrate the directed evolution (combined with rational approaches) of fungal laccases to facilitate their heterologous expression, extend or improve their catalytic activities, and adapt their enzymatic properties to specific conditions of application. The first directed evolution on a fungal laccase was carried out on a *M. thermophila* laccase toward functional expression in *Saccharomyces cerevisiae* [127]. Thereafter, the enzyme was evolved to enhance its catalytic activity in organic solvents [128]. Similarly, the functional expression in yeast of POXA1b laccase from *P. ostreatus* was addressed by combining error-prone PCR and DNA shuffling, whilst enzyme activity and stability were also enhanced [129,130]. The expression levels and catalytic constants of a laccase from *Fomes lignosus* (currently known as *Rigidoporus microporus*) were also enhanced by random mutagenesis [131].

The high-redox-potential laccases from the white-rot basidiomycetes PM1 and *Pycnoporus cinnabarinus* were subjected to directed evolution to improve their secretion by the yeast *S. cerevisiae* while improving their catalytic activities [132,133]. The coding sequences of the final evolved laccase variants from the two parallel evolutionary trajectories were thereafter randomly recombined by in vitro and in vivo DNA shuffling [134] to obtain a collection of chimeric laccases with modified pH activity profiles and substrate affinities and improved thermotolerances [135]. Next, the resulting random chimeric laccases were later used by our group as departure points for designing enzymes a la carte for specific applications. For instance, the re-design of the substrate-binding pocket of one of these laccases by iterative saturation mutagenesis (targeting six amino acid residues delimiting the pocket), allowed to improve the oxidation of natural phenolic compounds of biotechnological interest [15]. Additionally, the structured guided DNA recombination of PM1 and *P. cinnabarinus* evolved laccases [132,133] resulted in a domain-swap laccase with outstanding tolerance to high temperature and to the presence of organic co-solvents [136]. One major objective in enzyme engineering is the development of robust biocatalysts with improved activities towards specific substrates under the desired conditions of application. Through laccase-directed evolution assisted by computational design, our group has developed a robust enzyme that efficiently catalyzes the synthesis of conductive polyaniline structured in nanofibers and organic acid dyes for textile dyeing [10,137]. More recently, we have developed alkaliphilic and thermophilic high-redox-potential laccases for wood conversion processes. These evolved enzymes are able to boost kraft pulp bleaching, achieving 11% ClO_2_ savings; improve the de-fibering of wood chips in fiberboard production with less energy required; and depolymerize kraft lignins at pH 10 [11,85].

Other research studies have also addressed the in vitro evolution of laccases for industrial purposes. Because of the dependence of fungal laccase activity on acidic pH, the development of tailor-made laccases able to work at wider pH values is a recurrent engineering goal, together with providing activity under other non-natural conditions. For instance, the evolved PM1 laccase expressed in *S. cerevisiae* [132] was later subjected to directed evolution to make it active in human physiological fluids, shifting its pH-activity profile to more neutral values and improving its tolerance to halides [138]. In similar approaches, laccases from *T. versicolor*, *M. thermophila*, *Botrytis aclada* or *Cerrena unicolor* have been engineered to improve activity in ionic liquids [139], in broader pH ranges [140] or in higher pH and temperature [141,142], respectively. In the case of laccase Lcc9 from *Coprinopsis cinerea,* the optimal activity pH with phenol substrates was shifted to around pH 8 [143]. Additionally, directed evolution, using a fluorescence-activated droplet sorting system coupled with heat treatment, improved the resistance to organic solvents, ionic liquids and temperature of another fungal laccase [144].

### 4.2. Heterologous Expression

Laccases are produced by basidiomycete and ascomycete fungi, either constitutively or induced by the presence of lignin and aromatic compounds, copper, etc. [145,146]. Laccase secretion by saprotrophic basidiomycete species during lignocellulose decomposition has been reproduced in the laboratory by culturing the fungi under solid-state fermentation conditions on agro-industrial waste or wood chips. This approach provides valuable information on the potential use of these enzymes for the valorization of biomass waste [147,148]. 

In an attempt to boost the production of fungal laccases under controlled conditions, their homologous expression has been assayed in different studies [149,150,151]. The production yields offered by the homologous hosts are commonly insufficient for industrial purposes, although there are some remarkable exceptions, such as the overexpression of lac1 of *P. cinnabarinus* in a monokaryotic strain of this fungus under the regulation of the glyceraldehyde-3-phosphate dehydrogenase promoter, yielding up to 1.2 g/L [152]. However, basidiomycete fungi are not easily genetically manipulated, and the presence of multiple laccase gene isoforms may make the production and purification to homogeneity of a single targeted protein. Furthermore, not all fungi have a status of GRAS (Generally Recognized as Safe) organisms, making them incompatible with commercialization purposes. Therefore, heterologous expression is the best production alternative to achieve the efficient and simplified expression of these fungal enzymes.

The functional expression of fungal laccases in prokaryotic systems is notoriously difficult to achieve, so there are only a few reports on this topic. The laccase from the basidiomycete *Cyathus bulleri* became the first example of functional expression of a fungal laccase in the prokaryotic host (*E. coli*) [153]. However, laccase expression was only detected by zymogram, and no activity values were provided. Recently, the use of the Novel Signal Peptide 4 has contributed to the functional expression of a laccase from *T. versicolor* in *E. coli*, but the enzyme was secreted into the cell medium in much lesser amounts than other non-fungal laccases [154]. A possible explanation for the poor expression levels of fungal laccases in bacteria is related to differences between the host and the fungal post-translational modification machineries, which could lead to the formation of non-functional aggregates of the recombinant protein [155].

In contrast, filamentous ascomycete fungi are excellent hosts for fungal enzyme production. They often produce large quantities of proteins, far exceeding the capabilities of yeasts [156] (Figure 8). In addition, sugar anchoring by these organisms is more conservative than in yeasts, which tend to hyperglycosylate proteins [157], so downstream processing in filamentous fungi is easier. A first example of heterologous production of a white-rot laccase in *Aspergillus niger* is the production of lac1 from *P. cinnabarinus* at 70 mg/L [158], compared to the 8 mg/L of the same enzyme obtained in *Pichia pastoris* (current name, *Komagataella pastoris*) [159]. Subsequently, an evolved variant of the same *P. cinnabarinus* laccase was also produced in *Aspergillus niger*, resulting in 23 mg/L of the recombinant enzyme, a yield ten times higher than that obtained in *S. cerevisiae* [133]. The levels of a laccase from *Pycnoporus coccineus* produced in *Aspergillus niger* also far exceeded those obtained by natural expression [160], while the heterologous production of a *Trametes sp* laccase in the same filamentous fungus rendered remarkable production yields (up to 850 mg/L) [161]. *Aspergillus* is also a preferred heterologous host from an industrial point of view. For instance, *M. thermophila* laccase is produced at industrial scale in *Aspergillus oryzae* and is commercialized by Novozymes in different preparations. The recombinant enzyme was marketed under the trade names Flavourstar^®^ for food applications and DeniLite^®^ for denim fabric finishing. Different basidiomycete laccase variants engineered in our laboratory have been also overexpressed in *A. oryzae* at industrial relevant scale for the synthesis of conductive polyaniline [103] and of organic dyes whose textile dyeing properties were verified in an industrial environment [10]. In addition to *Aspergillus spp,* other filamentous fungi are also potential enzyme-producing hosts for the industry. For instance, the use of *Trichoderma reesei* as an expression system for laccase production provided 920 mg/L of *Melanocarpus albomyces* (ascomycete) laccase in fed-batch fermentation [162] and around 1000 mg/L of a *T. versicolor* laccase [163], whereas only 20 mg/L of *Phlebia radiata* laccase was obtained [164]. 

Despite the potential of filamentous fungi as enzyme-production systems, they are not easy to manipulate genetically for their use as hosts in directed evolution [165]. By contrast, yeast, and particularly *S. cerevisiae*, is considered a platform of choice for the directed evolution of fungal laccases. In addition to its easy genetic manipulation, the growth of *S. cerevisiae* is rapid and demands little, and it can perform the post-translational modifications required to secrete active eukaryotic enzymes into the culture broth [165,166]. In addition, the high frequency of the homologous recombination of *S. cerevisiae* [167] provides a crucial added value for the generation of mutant libraries of large sizes in enzyme engineering [127,168,169]. Moreover, laccase expression in *S. cerevisiae* can be enhanced through directed evolution, obtaining in some cases quite significant production levels (20–25 mg/L [10,127]), similar to those obtained in *P. pastoris*. The latter offers, in general, higher production yields than *S. cerevisiae* due to its high-density growth and less-glycosylated recombinant proteins [170]. Reported laccase production in this yeast yields range from 8 to 136 mg/L for *P. cinnabarinus* [159], *Trametes trogii* [171], *Trametes* sp. [172] and *B. aclada* [173] laccases, with the remarkable exception of 550 mg/L for *T. versicolor* laccase, produced in a fed-batch bioreactor using the AOX1 promoter [174]. Therefore, a recurrent strategy is to use a tandem-expression platform to engineer the enzymes in *S. cerevisiae* and over-express the evolved laccases in *P. pastoris* [11,175]. Nevertheless, yeast may show unpredictable behavior in expressing some coding sequences. For instance, differing results were obtained when laccases from *P. ostreatus* were produced in *S. cerevisiae* or *Kluyveromyces lactis* [156].

The native signal peptides of the wild-type enzymes are usually replaced by the signal peptides of most expressed proteins of the heterologous host due to their crucial role in protein expression [176]. The leader sequence of the α-factor mating pheromone of *S. cerevisiae* is among the most widely used leader sequences, known as α-factor prepro-leader. In terms of laccase expression, the fusion of this signal peptide to the enzyme has mainly contributed to the functional expression of these enzymes in both *P. pastoris* [172,177,178] and *S. cerevisiae* [10,127,132,133]. In fact, the recurrent use of the α-factor preproleader (fused to coding sequence of laccases) in the directed evolution campaigns has successfully enhanced enzyme secretion yields due to the accumulation of beneficial mutations in the evolved leader sequence. For example, the accumulation of five mutations in the α-factor preproleader during the directed evolution of *P. cinnabarinus* laccase in *S. cerevisiae* raised 40-fold laccase production compared to the native α-factor preproleader [133]. Thereafter, the highest laccase production levels ever reported for heterologous expression of basidiomycete laccases in *S. cerevisiae* (25 mg/L) was achieved with a further evolved α-factor preproleader (known as α9H2) [10]. Recently, we have studied the epistatic effects of mutations accumulated in this signal sequence through successive directed evolution campaigns and have been able to develop an optimized leader (αOPT) that markedly enhances the secretion of different fungal oxidoreductases and hydrolases [179].

## 5. Challenges and Opportunities

Actions to be carried out in the coming years within the European Bioeconomy Strategy and the Green Deal aligned with the UN Sustainable Development Goals seek to establish a bio-based economy in Europe, protect our natural habitats, and make Europe climate-neutral by 2050. The use of sustainably sourced plant biomass as a renewable raw material, and the development of environmentally friendly and circular industrial processes with reused or recyclable products, will contribute to meeting these goals. Industrial biotechnology is key to helping this transition to a green and circular economy. 

Oxidoreductase enzymes involved in lignocellulose biodegradation have great potential to contribute to attaining the integral use of plant biomass and waste as an alternative raw material to fossil feedstock to produce energy, chemicals and materials. However, the chemical industry has not yet embraced enzymatic oxidation reactions to a large extent. This is primarily due to the lack of biocatalysts, with the required activity and/or selectivity under the rigorous process conditions, that are available at reasonable scale for medium- and large-scale biotransformations. 

Fungal laccases have been traditionally described as biocatalysts with high applicability potential in different sectors. In recent years, the discovery and characterization of novel enzymes, the description of new redox mediators, the advances in protein engineering to obtain enzymes adapted to the required conditions of application, and the steady demonstration of their versatility to catalyze reactions of industrial and environmental interest (from chemical synthesis to removal of emerging pollutants in wastewaters) have demonstrated that laccases remain the subject of an active field of research. In recent years, laccases have gained renewed attention for the valorization of technical lignins from the pulp and paper and bioethanol industries due their ability to oxidize and deconstruct the lignin polymer. Due to their higher redox potential, laccases from ligninolytic fungi stand out as the preferred enzymes to realize the enzymatic depolymerization of leftover lignins into bio-based polymer building blocks (bioplastics included) and chemicals. However, the large-scale production of basidiomycete laccases is a pending challenge for the industrial implementation of these amazing enzymes. 

## Figures and Tables

**Figure 1 biomolecules-13-01716-f001:**
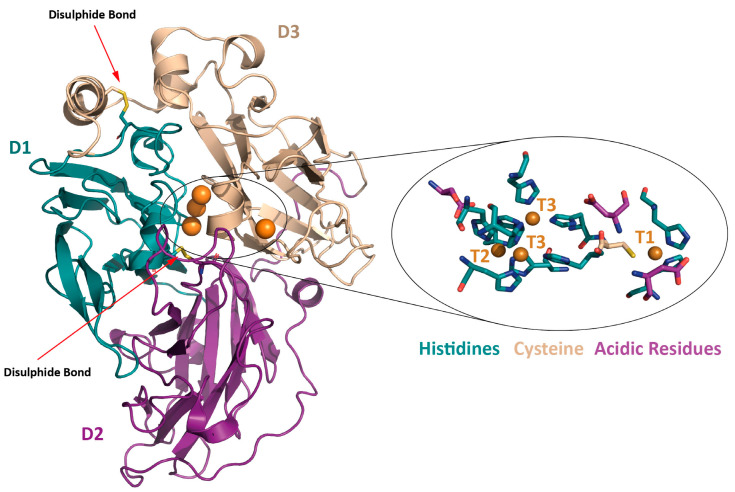
Cartoon representation of the crystal structure of the laccase from basidiomycete PM1 (*Coriolopsis* sp.) (PDB 5ANH, [15]) with the catalytic coppers depicted as orange spheres and the two disulphide bridges as yellow sticks. A zoom-up of the active site shows the histidine (in blue) and cysteine (in wheat) residues coordinating the four catalytic coppers and the conserved acidic residues assisting the catalysis (in purple).

**Figure 2 biomolecules-13-01716-f002:**
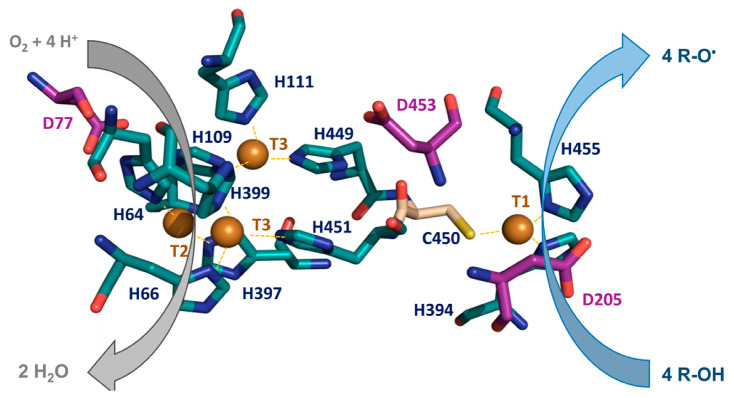
Electron transfer from the reducing substrate in the catalytic T1 site to TNC of laccases. The histidine (blue) and cysteine (wheat) residues coordinating the four catalytic coppers (depicted as spheres) and the conserved acidic residues involved in electron-proton transfer (purple) are shown in PM1 laccase structure (PDB: 5ANH) (adapted from De Salas and Camarero, 2021 [25]).

**Figure 3 biomolecules-13-01716-f003:**
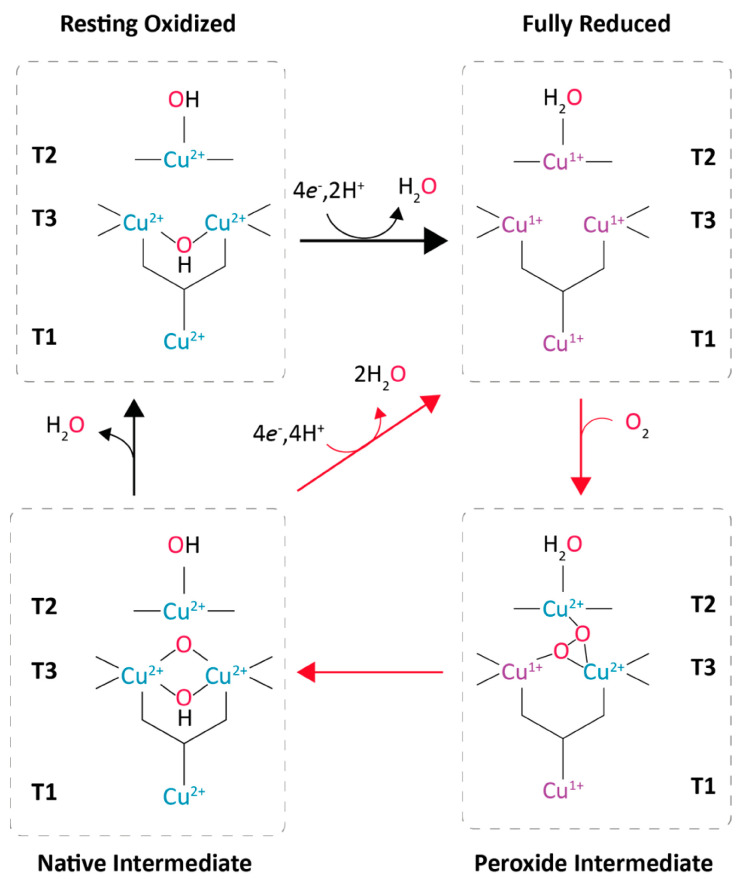
Reaction mechanism of laccases from the reduction of the T1 copper by substrate, the electron transference to the TNC and O_2_ reduction (adapted from Jones and Solomon et al., 2015 [17]).

**Figure 4 biomolecules-13-01716-f004:**
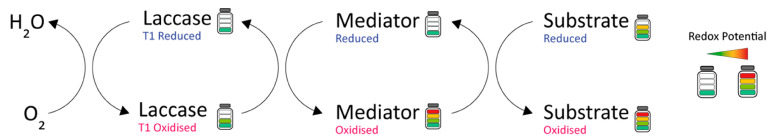
Laccase-mediator system for the oxidation of high-redox-potential substrates.

**Figure 5 biomolecules-13-01716-f005:**
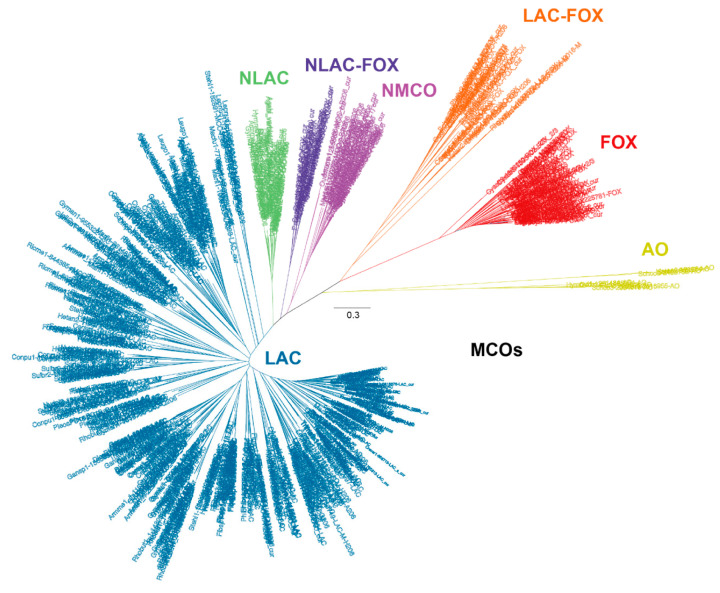
Phylogenetic analysis of the 649 MCO enzymes encoded in 52 basidiomycete genomes and classified as Laccase sensu stricto (LAC), Novel Laccase (NLAC), Novel MCO (NMCO), Novel Laccase-Ferroxidase (NLAC-FOX), Ferroxidase (FOX), Laccase-Ferroxidase (LAC-FOX) and Ascorbate Oxidase (AO). (Adapted from Ruiz-Dueñas et al., 2021 [63]).

**Figure 6 biomolecules-13-01716-f006:**
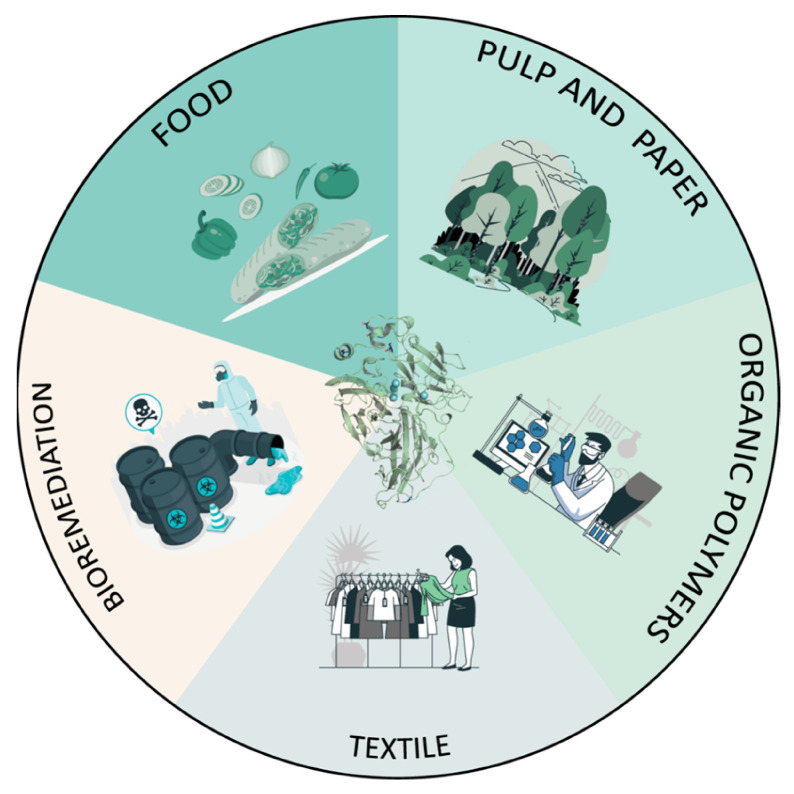
Sectors for the application of laccases as biocatalysts.

**Figure 7 biomolecules-13-01716-f007:**
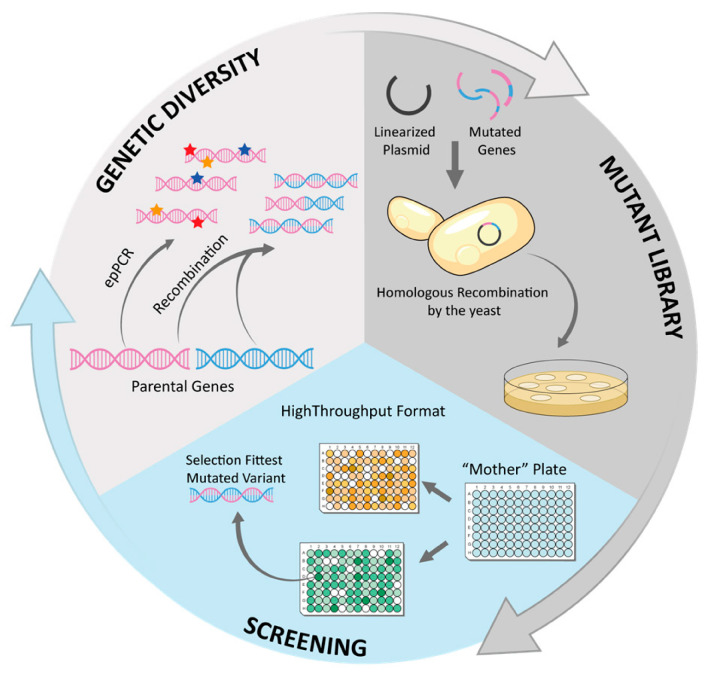
Scheme of a directed evolution cycle using yeast as an expression system.

**Figure 8 biomolecules-13-01716-f008:**
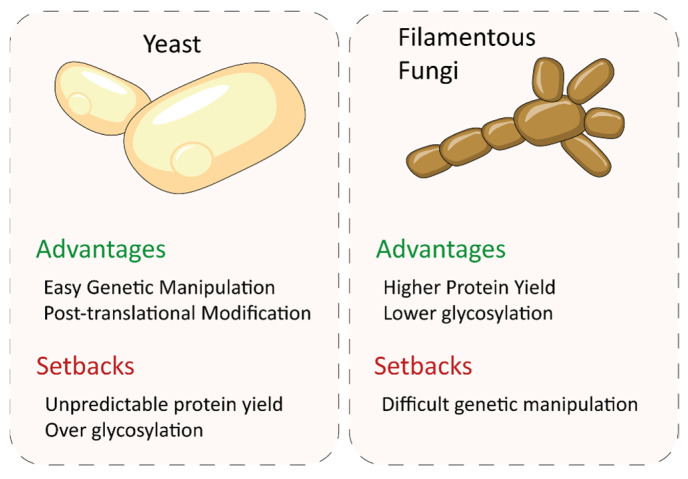
Comparison of yeasts and filamentous ascomycete fungi as expression systems of fungal laccases.

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
