# Peer review of "Fungal Laccases: Fundamentals, Engineering and Classification Update"

_biomolecules, 2023, doi:10.3390/biom13121716_

Round 1

Reviewer 1 Report

Comments and Suggestions for Authors

The main subject of the presented manuscript is fungal laccases. Although laccase is known for more than a century, it is still one of the most actively researched fungal enzyme worldwide. Typically, dozens of research articles and several reviews devoted to laccases are published each year; however, it is rare to encounter some conceptually new information and breakthrough ideas in these papers.

In my opinion, the main advantage of the presented manuscript is its conciseness. It successfully summarized all the main directions of the laccase research and still fits in 14 (not 140, as it could be!) pages. This manuscript definitely warrants the publication in the Biomolecules and especially in the “Recent Advances in Laccases and Laccase-Based Bioproducts” special Issue after some minor revisions.

My main concern regarding the presented manuscript is the lack of the references to the most current articles published in the field (almost all cited articles were published between 2000 and 2015). As I know, after Jones and Solomon et al., there were some modifications to the catalytic cycle of laccases made in 2017 (DOI:10.1107/S2059798317003667). Also, after the first attempts by Hoegger et al., which were made in pre-genomics era, the evolutionary classification of sensu stricto laccases were significantly improved in the 2019 (DOI:10.3389/fmicb.2019.00152). Hence, I suggest for authors to include the most current articles on the mentioned subjects into the reference list alongside the classical ones.

For the potential readers of the manuscript, it should be clear that "the science of laccases" is an active field of research and as more we know as more questions we get about this incredible enzyme.

Comments on the Quality of English Language

Although the grammar is good, some issues with style should be fixed. 

Author Response

Response: We very much appreciate the positive evaluation of the reviewer. As for his/her concern regarding the lack of references to the most recent articles, we acknowledge it and we have added the suggested references and several more to the revised version of the manuscript (highlighted in red).

In addition we have added a final section highlighting the still active research on laccases and future challenges and opportunities.

We have also carefully revised the English style.

Reviewer 2 Report

Comments and Suggestions for Authors

This is generally a well-written summary that focuses on the enzymology and biotechnological aspects of laccases and provides an extensive review of the literature. There are places where the English is a little rough (see below for examples). 

Major comments

The manuscript would be greatly strengthened by adding a summary paragraph and ideally some forward-looking comments on the most urgent challenges and largest potential opportunities for using laccases. 

Plant and bacterial laccases are mentioned in the introduction but almost completely ignored throughout the manuscript. The authors should either provide a explicit rationale for focusing on fungal laccases or provide some brief treatment of laccases from other organisms. 

detailed comments

I would suggest moving the reclassification section (4. Multicopper oxidases reclassification) to directly after section 1 as this section builds up on the laccase structure and activity presented in section 1

The Biotechnological applications section is short. The section (and especially the organic synthesis section) could use more detailed examples of how laccase are used and what the advantages are. And I feel there are probably more sectors in which laccases are used, for example, in biomass breakdown for biofuel production.

14 Unclear wording...maybe  "low-sequence homology (makes it) difficult (to accurately) classify (them)?"

22 should be 'newly' vs 'new'

85-102 it seems that T1 needs to be involved in the 4 electron process and yet this is not clear in the current description. 

85 provide a reference for the battery analogy...though I'm not sure that's the best because a battery separates charges. A capacitor stores charge. 

The sentence starting on 126 is confusing

It would be nice to add some description of the advantages and disadvantages of high vs. low potential laccases

155 Unclear what you mean by 'molecules that are not easily oxidised by laccases' please be more specific (e.g. do you mean they need a higher redox potential?)

194 Unclear how oxidative activity towards indigo-dyes results in abrasion effects

245 should be 'early' not 'earliest'

paragraph starting on line 268, describe how the selection works (e.g. enzyme activity assays, microbial selection, etc)

346 change 'as regards filamentous' to "In the case of ascomycete fungi" or something like this

Comments on the Quality of English Language

See  my other comments on the writting 

Author Response

REVIEWER 2

Comments and Suggestions for Authors

This is generally a well-written summary that focuses on the enzymology and biotechnological aspects of laccases and provides an extensive review of the literature. There are places where the English is a little rough (see below for examples).

Response: thank you for your positive comment. Regarding the English style, we have carefully revised it throughout the text.

Major comments

The manuscript would be greatly strengthened by adding a summary paragraph and ideally some forward-looking comments on the most urgent challenges and largest potential opportunities for using laccases.

Response: we have added a new section (5) where we give a brief outlook of the most important challenges and opportunities for the application of these enzymes in the context of a Circular Bioeconomy, as well as the main limitations.

Plant and bacterial laccases are mentioned in the introduction but almost completely ignored throughout the manuscript. The authors should either provide a explicit rationale for focusing on fungal laccases or provide some brief treatment of laccases from other organisms.

Response: is true we give a very limited coverage to bacterial and plant laccases. However, as indicated in the title, we aim to focus on fungal laccases, and particularly on those secreted by ligninolytic basidiomycete fungi. Following the reviewer’s suggestion, the rationale for this decision is better explained now in the revised text (see revised abstract and lines 40-49 at the beginning of the Introduction section).

detailed comments

I would suggest moving the reclassification section (4. Multicopper oxidases reclassification) to directly after section 1 as this section builds up on the laccase structure and activity presented in section 1

Response: done

The Biotechnological applications section is short. The section (and especially the organic synthesis section) could use more detailed examples of how laccase are used and what the advantages are. And I feel there are probably more sectors in which laccases are used, for example, in biomass breakdown for biofuel production.

Response: we agree the applications section is short. However, as explained now in the last paragraph of first section, we aimed to focus on other relevant aspects of fungal laccases such as the recent advances on laccase engineering and classification. Nevertheless, according to the reviewer’s comment, we have added some more details on recent studies on the use of laccases in organic synthesis (see lines 391-395 and 401-405).

14 Unclear wording...maybe "low-sequence homology (makes it) difficult (to accurately) classify (them)?"

Response: the sentence has been rewritten as follows: “On the other hand, their substrate versatility and the low sequence homology among laccases make their exact classification difficult.”

22 should be 'newly' vs 'new'

Done

85-102 it seems that T1 needs to be involved in the 4 electron process and yet this is not clear in the current description.

85 provide a reference for the battery analogy...though I'm not sure that's the best because a battery separates charges. A capacitor stores charge.

Response: we appreciate the two comments and agree that “battery” analogy may lead to confusion, so we have remove this term. Please see the amended paragraphs (lines 90-92 and 102-109) rewritten to avoid confusion.

The sentence starting on 126 is confusing

Response: we have deleted the sentence

It would be nice to add some description of the advantages and disadvantages of high vs. low potential laccases

Response: we have added an explanation at the beginning of section 1 (lines 40-49).

155 Unclear what you mean by 'molecules that are not easily oxidised by laccases' please be more specific (e.g. do you mean they need a higher redox potential?)

Response: the reviewer is right, the sentence refers to those molecules whose high redox potential hinder their direct oxidation by laccase, and also to complex substrates that are not readily accessible to the enzyme binding pocket. We have rewritten the paragraph to better explain this matter (lines 161-165).

194 Unclear how oxidative activity towards indigo-dyes results in abrasion effects

Response: we have removed “abrasion” because we agree it can be confusing. We meant worn or faded appearance, usually desired in denim textile. See line 365.

245 should be 'early' not 'earliest'

Response: done

paragraph starting on line 268, describe how the selection works (e.g. enzyme activity assays, microbial selection, etc)

Response: following the suggestion of the reviewer, we have added some minor details on the selection of best mutants using high-throughput screening methods. We don’t intend to thoroughly describe all different screening or selection methods available for directed evolution, which have been addressed in other article reviews.

346 change 'as regards filamentous' to "In the case of ascomycete fungi" or something like this

Response: done
